# Mineral Composition of Skeletal Elements in Dorid Nudibranchia *Onchidoris muricata* (Gastropoda, Mollusca)

**DOI:** 10.3390/biomimetics10040211

**Published:** 2025-03-29

**Authors:** Dmitry A. Ozerov, Ekaterina D. Nikitenko, Alexey A. Piryazev, Andrey I. Lavrov, Elena V. Vortsepneva

**Affiliations:** 1N.A. Pertsov White Sea Biological Station, Faculty of Biology, Lomonosov Moscow State University, Leninskie Gory 1-12, 119234 Moscow, Russia; ozdm@ya.ru (D.A.O.); nikitenkocatia@yandex.ru (E.D.N.); lavrovai@my.msu.ru (A.I.L.); 2Russian Academy of Sciences Federal Research Center of Problems of Chemical Physics and Medicinal Chemistry (FRC PCPMC RAS), Ac. Semenov Avenue 1, Chernogolovka, 142432 Moscow, Russia; stunnn@gmail.com; 3Department of Chemistry, Lomonosov Moscow State University, 119991 Moscow, Russia; 4Invertebrate Zoology Department, Faculty of Biology, Lomonosov Moscow State University, Leninskie Gory 1-12, 119234 Moscow, Russia

**Keywords:** EDX, calcite, spicule, biomineralization, compositional data analysis, CoDA

## Abstract

Energy-dispersive X-ray spectrometry (EDX), a standard technique in mineralogy and criminalistics, has not yet been fully incorporated into the study of various biomineral structures of invertebrates, despite the growing popularity of this topic in the last few decades. This is partly due to EDX’s limitations and data interpretation complexities. This study used EDX to analyze the spicules’ elemental composition of nudibranch gastropod mollusk *Onchidoris muricata* prepared via two methods (sectioning and fracturing). Hierarchical clustering and compositional data analysis of the resulting elemental data revealed three distinct spicule populations with varying element ratios, suggesting spicule transformation pathways. Two of the three clusters had a uniform layered microstructure, yet they showed reliable differences in element ratios. Raman spectroscopy confirmed the spicules’ calcite or magnesian–calcite composition. EDX analysis of spicule sections, coupled with other analytical techniques, revealed mineral structure transformations and provided insights into the biomineral nature. The sample preparation method with epoxy-embedding, preserving surrounding tissues in their active state, allowed for the analysis of tissue elemental composition and the elucidation of their role in mineralization.

## 1. Introduction

Spicules are skeletal elements widespread in living organisms, from unicellular organisms to chordates [1,2,3,4,5,6,7,8,9,10,11]. Spicules are formed through biomineralization—the controlled creation of mineral structures based on an organic matrix in biological systems [12,13,14]. Studying the mechanisms and regulation of biomineralization is crucial for understanding the evolution of skeletal structures and developing novel biomaterials and biotechnologies. Understanding spicule structural, chemical, and physical properties, architecture, and composition will inform the design of biomimetic materials. These spicule-inspired materials hold promise for applications in materials science, biomedicine, and tissue engineering, such as bone regeneration scaffolds. While spicules are diverse, detailed studies of their morphology, formation, mineralization, and regeneration are primarily limited to model or commercially important species (e.g., certain sponges and sea urchins) [4,15,16,17,18,19,20,21,22,23,24].

The spicules of nudibranch gastropods (Doridina, Nudibranchia) are unique [25,26], differing fundamentally in morphology and formation from other invertebrate skeletal elements due to their intracellular location within vacuoles, unlike the extracellular formation seen in worm-like aplacophoran mollusks [27,28]. This makes nudibranch spicules ideal models for studying intracellular biomineralization. Their formation, growth, and transformation within sclerocytes involve internal structural changes, though the precise mechanisms remain unclear.

Current biomineralization research employs various approaches focusing on the organic matrix [4,11,18,24,29,30], relevant genes [16,17,18,19,20,21,22], and mineral identification [17,19,27,28,31]. Determining elemental composition offers additional insights into the spicule nature. Energy-dispersive X-ray spectroscopy (EDX), widely used in geology [32,33,34,35], is increasingly applied in biology [36,37,38,39,40,41,42,43,44,45,46], but the heterogeneous, porous structure of biominerals due to organic components requires methodological adaptations and rigorous statistical analysis for accurate interpretation [43,44,47].

This study used the well-characterized spicules of *Onchidoris muricata* (Doridina, Nudibranchia) to investigate the impact of sample preparation on EDX analysis. *Onchidoris muricata* spicules form a dense layer beneath the integument (Figure 1). While their morphology and ontogeny are well-understood [26,48], their chemical composition remains unknown, with the prior literature only mentioning the potential presence of calcium or magnesium carbonate, as well as of magnesium and organic impurities [49,50,51,52].

This work addresses whether sample preparation (using either fractures of dried specimens or epoxy resin-embedded sections) affects the accuracy of elemental composition analysis. The latter method provides a smooth, non-porous surface, crucial for accurate EDX analysis as, according to the established literature [53], surface irregularities can obstruct characteristic radiation. Raman spectroscopy, which is capable of analyzing both organic and mineral components and identifying specific mineral isoforms, was used to validate the EDX data.

## 2. Materials and Methods

### 2.1. Materials

Specimens of *O. muricata* were collected in the vicinity of the N.A. Pertzov White Sea Biological Station of Moscow State University in the Kandalaksha Bay, the White Sea (WSBS; 66°34′ N, 33°08′ E). To study spicules using EDX, 6 specimens ranging in size from 3 to 8 mm were collected during the summer–autumn seasons of 2022–2024. For Raman spectroscopy analysis, spicules from 15 specimens of various sizes were examined.

### 2.2. Sample Preparation for EDX

To make polished sections, the samples were embedded in resin using a standard preparation method for TEM [26]. To study fractures, the fixed samples were dehydrated and dried at a critical point following standard SEM sample preparation techniques [26]. In total, two specimens (Om27 and Om37) were embedded in resin; fractures were prepared from two other specimens (Om4 and Om8); one specimen (Om38) was sagittally sectioned into two halves, one of which was embedded in resin while the other was fractured. EDX analysis was performed on 100 spicules from sections and 105 spicules from fractures.

The initial steps of sample preparation were the same for both methods. Before fixation, *O. muricata* individuals were relaxed in a magnesium chloride (MgCl_2_·6H_2_O) solution isotonic to seawater at 8–10 °C for 1–4 h. The samples were then fixed in 2.5% glutaraldehyde (GA) in phosphate buffer for at least 2 h at 4 °C, with the fixative changed after 1 h. After fixation, the specimens were washed three times with the same buffer for 20–25 min each time and postfixed in 1% OsO_4_ in the same buffer for 1 h at room temperature in the darkness. The samples were then washed with the buffer and dehydrated in a series of increasing concentrations of ethyl alcohol (30%–50%–70%–85%–96%), followed by a sequence of ethanol–acetone mixtures (3:1–1:1–1:3–acetone–acetone) for 25–30 min each at room temperature with shaking.

For fracture analysis, the dehydrated samples were dried at a critical point. Fractures were then made from dried fragments using steel blades.

For sectioning, specimens were embedded in Spurr resin (Sigma-Aldrich, St. Louis, MO, USA) without decalcification. The samples were successively placed in three acetone–Spurr resin mixtures (3:1–1:1–1:3) at room temperature with shaking for 24 h in each mixture. Next, they were placed in pure Spurr resin I overnight in a refrigerator with an open lid to allow residual acetone to evaporate. The samples were then transferred to resin II in molds at 37 °C overnight. Polymerization was carried out at 60 °C for 24 h. After polymerization, the blocks with embedded samples were sectioned by grinding and polishing.

Grinding and Polishing. The embedded material was oriented so that the section passed transversely through the anterior third of the body. Initial grinding was performed using a coarse file. Then, the samples were polished using aluminum oxide abrasives ranging from P400 to P7000 grit, followed by fine polishing. Fine polishing was carried out using two methods: a sequence of diamond pastes with decreasing grain size and cerium oxide powder. The best results were achieved using ZM-525 Platinum (Zauber Machinery, Moscow, Russia) cerium oxide powder (average particle size, 1.0–1.1 µm) designed for glass polishing. The single-stage cerium oxide polishing was faster than the multi-stage diamond paste method and minimized spicule embedding, eliminating the need for a subsequent chamois polishing step that could worsen this issue. Additionally, cerium oxide avoided sample contamination from the binder present in diamond pastes. Polishing was performed using both a portable disk sander and manual methods. The highest levels of control, predictability, and quality were achieved with manual processing, which was used for all analyzed sections. After polishing, the samples were washed with a surfactant and then placed in an ultrasonic bath in distilled water for 5 min.

### 2.3. EDX Analysis

For EDX analysis, the samples were coated with a 585 gold alloy containing small amounts of silver and copper using an Eiko IB3 ion sputter coater (EIKO Corporation, Tokyo, Japan) for 7–15 min. Data collection from both sections and fractures was performed using a JEOL JCM-7000 SEM (JEOL, Tokyo, Japan). Imaging was conducted under a high vacuum at a 15 kV accelerating voltage, with a working distance of 12 mm and a real-time parameter of 90 s. Internal spicule analysis focused on areas with a uniform internal structure, which were visually assessed. The obtained spectra were analyzed in Smile View Lab v. 3.8.3.0 with EDS components v. 3.8.3.0 (JEOL). Element identification was performed automatically, with subsequent manual verification according to the energy table for EDX analysis. The elements present in the sputter coat (Au, Cu, Ag) were excluded from the quantitative analysis. Intensities of characteristic X-rays were converted to relative quantities of elements through ZAF correction in the standardless mode with internal normalization to 100%. The relative quantities were presented as proportions of the number of atoms of each element.

### 2.4. Statistical Analysis

Statistical analysis was performed in R Statistical Software (ver. 4.4.1) with the basic packages “stats” ver. 4.4.1 and “graphics” ver. 4.4.1 (R Core Team, 2024) and additional packages “dplyr” ver. 1.1.4 [54], “easyCODA” ver. 0.40.2 [55], “compositions” ver. 2.0-8 [56], “factoextra” ver. 1.0.7 [57], and “emmeans” ver. 1.10.6 [58]. Full R scripts and data frames are available in the Appendix A.

Relative quantities of the elements in spicules were treated as compositions, and statistical analysis was performed according to Aitchison’s [59] concept of compositional data analysis, i.e., the data were analyzed in a log-ratio-transformed scale. To avoid zero values, low-represented elements (Cl, F, P) were amalgamated into a single part, “traces”.

Elucidation of a possible grouping of the samples was performed through an agglomerative hierarchical clustering. The clustering was performed on the centered log-ratio (CLR)-transformed data using Euclidian weighted log-ratio distances and Ward’s linkage method [55]. A central tendency of the obtained clusters was described using centers (compositional mean) and total (metric) variance as a measure of the global data spread [60]. To evaluate differences in compositions between clusters, a linear regression model with composition as a dependent variable was built using isometric log-ratio (ILR)-transformed data [60]. The global response and pairwise comparisons were assessed. The codependencies of the elements were analyzed in each cluster through pairwise log ratios, variation matrices, and biplots [60,61].

### 2.5. Raman Spectroscopy

In addition to EDX analysis, *O. muricata* spicules were analyzed using Raman spectroscopy to determine their chemical composition. For analysis, body fragments of *O. muricata* were placed in a 10% sodium hypochlorite (NaOCl) solution on a glass slide with a depression for 1–3 min at room temperature. The dissolution of soft tissues was visually monitored using a Nikon binocular microscope (Nikon Corporation, Tokyo, Japan). Once soft tissues began to dissolve, the body fragment containing spicules was transferred to distilled water. Washing with distilled water was performed in five changes, each lasting 5–10 min. Individual spicules were extracted from the soft tissue using fine needles and transferred onto a glass slide in a drop of water. The samples were left to air-dry at room temperature until the water evaporated completely.

Raman spectra were acquired using Olympus U-CTR30-2 (Olympus, Tokyo, Japan) and a Horiba LabRam Evolution Raman spectrometer (Horiba, Longjumeau, France), equipped with a 532 nm wavelength laser. The spectrometers were equipped with two diffraction gratings, a high-performance detector cooled by a Peltier element, and a set of neutral gray filters to adjust the radiation power on the sample. The optical microscopes had a resolution of 500 nm. The measurements were carried out with a grating of 600 lines per mm, the laser power was 10% of the maximum, and the objective was ×50. The exposure time was 10 s with two accumulations.

## 3. Results

### 3.1. Internal Morphology of Spicules

Examination of the *Onchidoris muricata* sections revealed two types of spicules (Figure 1). Layered spicules were the dominant type in the body wall, with varying sizes of up to 50 μm in diameter (Figure 1D,E). These spicules featured a cortical layer and numerous internal concentric layers in the transversal sections. The cortical layer contained pores into which resin penetrated, causing the appearance of a radial structure on the section. Homogenous spicules comprised the second type, with diameters of up to 30–40 μm (Figure 1C). They were rarely detected in thin sections of the body wall. The study of the morphology of spicules on fractures also revealed two types of internal structures. However, homogenous spicules were the predominant type on fractures (Figure 1F,G), while layered spicules were occasional (Figure 1H). The obtained data indicate the limitations of morphological methods for identifying spicule types and the importance of elaborating a standard sample treatment to compare results across multiple experiments.

### 3.2. Analysis of Spicule Elemental Composition Using EDX

As EDX analysis is recommended to be performed on a polished surface to avoid distortions of X-ray emission due to local topographic effects, the main analysis of the elemental composition of *O. muricata* spicules was conducted on the sections. A total of 100 spicules from sections and 105 spicules from fractures were studied. Four main elements, C (carbon), O (oxygen), Ca (calcium), and 1.6% Mg (magnesium), were detected in all the spicules.

### 3.3. Grouping of Spicules According to the Mineral Composition Based on Sections

Agglomerative hierarchical clustering revealed three clusters of spicules differing in the ratios of elements (Figure 2).

The first cluster (SC I) included 14 small morphologically homogeneous spicules. These spicules differed greatly from spicules of the other two clusters in both morphology and mineral composition. The SC I spicules were characterized by the following mean composition: 20.07%—C, 46.04%—O, 19.8%—Ca, 1.6%—Mg, and a fairly high proportion of trace elements, 12.4% (Table 1, Figure 3). The high proportion of trace elements arose due to the occurrence of P (phosphorus) in the SC I spicules, which is a unique characteristic of their mineral composition in comparison with spicules of the other clusters (Appendix A).

The spicules of the second cluster (SC II) were layered. A total of 41 spicules were united in this cluster. The mean composition of the SC II spicules was as follows: 9.46%—C, 61.96%—O, 21.73%—Ca, 3.93%—Mg, and 2.9%—traces. In comparison with the SC I spicules, the SC II spicules contained approximately the same amounts of Ca and Mg, more O, and less C and trace elements (Table 1, Figure 3). It is noteworthy that P was completely absent in the SC II spicules (Appendix A).

The spicules of the third cluster (SC III) were layered and morphologically indistinguishable from the SC II spicules. A total of 45 spicules were united in SC III. The mean composition of the SC III spicules was as follows: 8.06%—C, 53.5%—O, 33.82%—Ca, 2.88%—Mg, and 1.74%—traces (Table 1, Figure 3). The SC III spicules were comparable to the SC II spicules in the C, Mg, and trace element content, but had a higher amount of Ca and a lower amount of O; the SC III spicules also contained no P (Appendix A). The distinguishing characteristic of the mineral composition of the SC III spicules was their high content of Ca.

For all clusters, the total variance did not exceed 0.2567 (Table 1), indicating low within-cluster data variability and general compactness of the clusters.

Regression analysis indicated that the mineral composition of spicules was significantly different between the clusters (ANOVA *p*-value < 0.0001) (Appendix A). Subsequent pairwise comparisons confirmed that all clusters significantly differed from each other in mean mineral composition (*p*-values < 0.0001 for each pair of clusters) (Appendix A).

### 3.4. Analysis of Element Ratios in the Clusters

The core differences in mineral composition between the clusters were assessed by analyzing the pairwise ratios of the elements (Figure 4; Table 2).

SC I differs significantly in the element ratios from SC II and SC III: the SC I spicules are clearly separate from the SC II and SC III spicules on most ternary diagrams (Figure 4A). SC I is characterized by low ratios of each element to the traces, of each element to C, and of Mg to O (Figure 4B; Table 2). These ratios showed low values in SC I (Table 2) due to the higher content of traces and C and the lower content of Mg in the spicules of this cluster (Table 1). In contrast, SC II and SC III did not differ from each other in the majority of these ratios (Mg/O, Mg/C, O/C, Mg/Tr, O/Tr, C/Tr) (Figure 4B; Table 2), and spicules from these clusters intermingled on the corresponding ternary diagrams (Figure 4A). The main differences between SC II and SC III lay in the ratios involving Ca: SC III had high ratios of Ca to each element (Figure 4B; Table 2) due to the higher content of Ca (Table 1). Notably, SC I and SC III demonstrated similar Ca/Mg ratios, while in SC II, this ratio was lower (Figure 4B; Table 2).

Despite the described differences, the variation matrices for each cluster showed fairly low values (not exceeding 0.2370; Table 3), pointing to the quasi-constancy of all ratios in each cluster.

### 3.5. Grouping of Spicules According to the Mineral Composition Based on Fractures

To assess the correspondence of the data on the elemental composition of spicules prepared in different ways, we additionally analyzed spicules on fractures (Appendix A). Agglomerative hierarchical clustering revealed four clusters of spicules (Figure 5). The clusters identified on the fractures did not clearly correspond to the clusters from the sections.

The first spicule fracture cluster (FC I) included 19 spicules, which were characterized by the following mean composition: C—14.79%, O—29.69%, Ca—36.53%, Mg—1.26%, and also a fairly high percentage of trace elements, 17.73% (Table 1, Figure 5). The traces included P, Cl, and F (Appendix A). The second spicule fracture cluster (FC II) included 27 spicules, which were characterized by the following mean composition: C—11.58%, O—53.55%, Ca—29.95%, Mg—3.20%, and also a fairly low percentage of trace elements, 2.07% (Table 1, Figure 5). The traces also included P, Cl, and F, but P was detected only in 2 spicules (Appendix A). The third spicule fracture cluster (FC III) included 19 spicules, which were characterized by the following mean composition: C—10.46%, O—62.46%, Ca—16.75%, Mg—5.32%, and also a rather low percentage of trace elements, 5% (Table 1, Figure 5) (P, Cl, F). P was detected only in 5 spicules (Appendix A). The fourth spicule fracture cluster (FC IV) included 40 spicules, which were characterized by the following mean composition: C—17.31%, O—53.32%, Ca—15.91%, Mg—1.91%, and a high percentage of trace elements, 11.54% (Table 1, Figure 5) (P, Cl, F), and P was detected in all spicules except one (Appendix A). The total variance of the fracture clusters was higher in comparison with the section clusters (Table 1). Accordingly, pairwise element ratios showed considerable ranges (Table 2), and the corresponding values in variation matrices were rather high (Table 3). All these values indicated heterogeneity of the spicules within the fracture clusters and thus sparse clustering.

Further hierarchical analysis of data on spicules from the Om38 specimen (Appendix A), half of which was polished and the other half fractured, confirmed the heterogeneity of the data obtained for the sections and fractures. In general, the spicule sections were clustered separately from the spicule fractures: cluster 1 combined all spicule sections and two spicule fractures, while clusters 2 and 3 grouped the rest of the spicule fractures (Figure 6). This result likely indicates a strong batch effect due to the different types of sample processing. Thus, simultaneous analysis of the data obtained with different sample preparation methods seems to be impossible.

### 3.6. Raman Spectroscopy Data

The chemical composition of the *O. muricata* spicules was studied using Raman spectroscopy, which significantly complements the EDX method. For most spicules, the Raman shift was characterized by the presence of three characteristic peaks at 281–283 cm^−1^, 707–720 cm^−1^, and 1087–1088 cm^−1^ (symmetric stretching mode of the carbonate ion) (Figure 7 and Appendix A), which corresponded to the characteristic peaks of biogenic calcite and magnesian calcite [17,19,31]. In some spicules, the characteristic peaks at 281 cm^−1^ or 700 cm^−1^ were not expressed.

## 4. Discussion

### 4.1. Methodological Aspects

This study aimed to find the optimal sample preparation protocol for *Onchidoris muricata* spicules to enable reliable elemental analysis of spicules that would allow conducting not only qualitative, but also comparative quantitative analysis of elements in samples. Analysis of the spicules on fractures revealed a higher variability (total variance) of elements compared to the sections. This is attributed to the uneven surface of fractures and their difficulty in precise horizontal orientation relative to the detector, both factors compromising data objectivity. Hierarchical cluster analysis further supported this, showing a clear separation of the spicule sections versus the spicule fractures. Based on the data from this work, as well as on earlier articles [41,42,43,44,45], we can conclude that it is necessary to make sections to evaluate quantitative elemental analysis. Fractures should be used only for a preliminary assessment of the qualitative composition of the structures.

A potential key advantage of resin-embedded sections is the preservation of surrounding soft tissues involved in biomineralization, allowing EDX analysis of these tissues to illuminate formation mechanisms. Furthermore, the sections provided a clearer spicule microstructure. A comparison of the same spicules in a single specimen (Om38) identified differences between the fracture and section preparations: the fractures showed predominantly homogeneous spicules (consistent with previous findings), while the sections revealed a layered structure, indicating that spicules’ internal morphology is strongly influenced by sample processing.

### 4.2. Analysis of Spicule Mineral Composition

Elemental analysis of the *O. muricata* spicule sections revealed three significantly different spicule clusters (SC I–III). SC I was distinguished by high carbon and trace element content, while SC II and SC III were characterized by low carbon/trace element content; SC III additionally showed the highest calcium levels. The low variability in element ratios across all clusters suggests a strong inter-element relationship, possibly indicating that they comprise a single mineral/compound.

Based on the earlier morphological studies, presumed stages of *O. muricata* spicule maturation were proposed as follows: an organic matrix inside the sclerocyte vacuole gradually mineralized through stages with a concentric structure to a completely monolithic large spicule. The clusters identified in this study based on EDX data likely represented different stages of spicule maturation.

The use of grinding sectioning in this study indicates that large, fully monolithic spicules are absent in *O. muricata*. Previously identified large “monolithic” spicules possibly have a dense concentric structure, which was masked on fractures. The SC I spicules seemed to be truly monolithic but were smaller than the other spicules. Moreover, they contained less calcium, but more carbon and trace elements, notably phosphorus (Figure 4). The SC II and SC III spicules (Figure 2A) showed morphological similarity as both clusters contained layered spicules. The layered appearance of the mineral structure is not unique to *Onchidoris*: deposition of minerals concentrically around an axis has also been described in sponges, and it is likely the most common variant of mineral deposition in biomineralization processes [16,17,62,63,64]. In this case, layers of minerals alternated with organic thin interlayers. Spicules of both the SC II and SC III clusters contained more calcium than the SC I spicules, which may indicate that the SC II and SC III spicules represented more advanced stages of the mineralization process. SC II and SC III differed from each other mainly in the ratios of Ca to the other elements: the SC III spicules contained more calcium and were probably more mature.

Previous studies concerning a few dorid species showed a confusing picture of the mineral composition of spicules [27,28,50]. Spicules could contain different isoforms of calcium carbonate (amorphous calcite [50], calcite, vaterite [51], monohydrocalcite [65]), brucite (Mg(OH)_2_), fluorite (CaF_2_) [51], and amorphous fluorite [66]. The mineral composition of *O. muricata* spicules was characterized for the first time. Consistent with Raman spectroscopy, all three clusters contained calcite and magnesian calcite (Figure 7); no other minerals were identified (Appendix A). At the same time, quantitative analysis of the elemental composition revealed variations in the Ca/Mg ratio among the different spicules. We hypothesize that this may be due to the substitution of calcite with magnesium calcite in various spicules, potentially influenced by factors such as the degree of maturation, individual variations, etc. This corresponds to the observation that calcium is often replaced by magnesium within the crystal lattice of biogenic carbonates, potentially influenced by water temperature and ocean saturation [31].

A notable finding is the consistent detection of magnesium in all the analyzed spicules, including SC I, suggesting its presence extends beyond magnesian calcite. The literature indicates the crucial role of magnesium in stabilizing the organic matrix of biominerals [19], a function likely relevant to *O. muricata* spicules. Previous studies also mentioned that Mg^2+^ and (PO_4_)^3–^ can adsorb onto the surface of a mineralized structure or be incorporated into the structure of amorphous calcium carbonate and slow down its crystallization [66].

## 5. Conclusions

For reliable elemental analysis of spicules and surrounding tissues, resin embedding and polishing are essential; fractured samples are suitable only for preliminary assessment. Rigorous statistical analysis is crucial for the accurate interpretation of EDX data: this method generates data with a sum constraint representing only relative values of elements. Such datasets are unsuitable for the direct implementation of classical statistical approaches. Compositional data analysis (CoDA) is an appropriate statistical approach for EDX data analysis as it eliminates mathematical paradoxes by considering not the original values of the elements, but rather their ratios.

Our data showed high correlations between all elements in all the spicules, suggesting a single system undergoing continuous compositional transformations. However, three spicule clusters were identified: a homogeneous (likely initial) stage and two layered stages. Intriguingly, these layered groups, while morphologically indistinguishable, exhibited distinct element ratios. Combined with other methods and proper statistical analysis, EDX provides not only qualitative elemental composition data, but also, through element ratios, insights into ontogenetic mineral structure transformations and the characteristics of morphologically similar developmental stages.

## Figures and Tables

**Figure 1 biomimetics-10-00211-f001:**
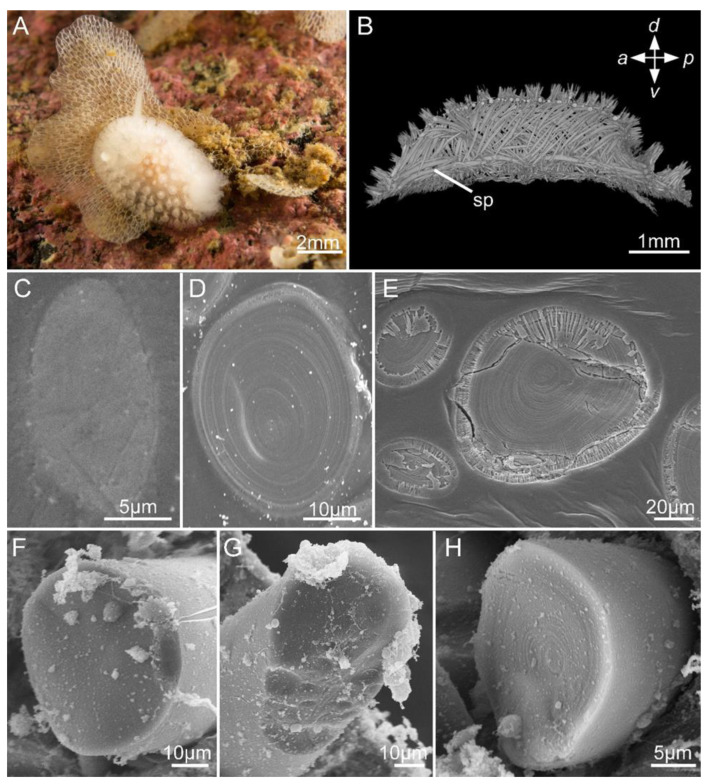
Internal morphology of *Onchidoris muricata* spicules. (**A**) A mollusk in its natural habitat. (**B**) Organization of spicule tracts in the body, microCT, lateral view. (**C**–**H**) Internal spicule morphology. Data obtained from sections (**C**–**E**) and fractures (**F**–**H**) Note: sp, spicule.

**Figure 2 biomimetics-10-00211-f002:**
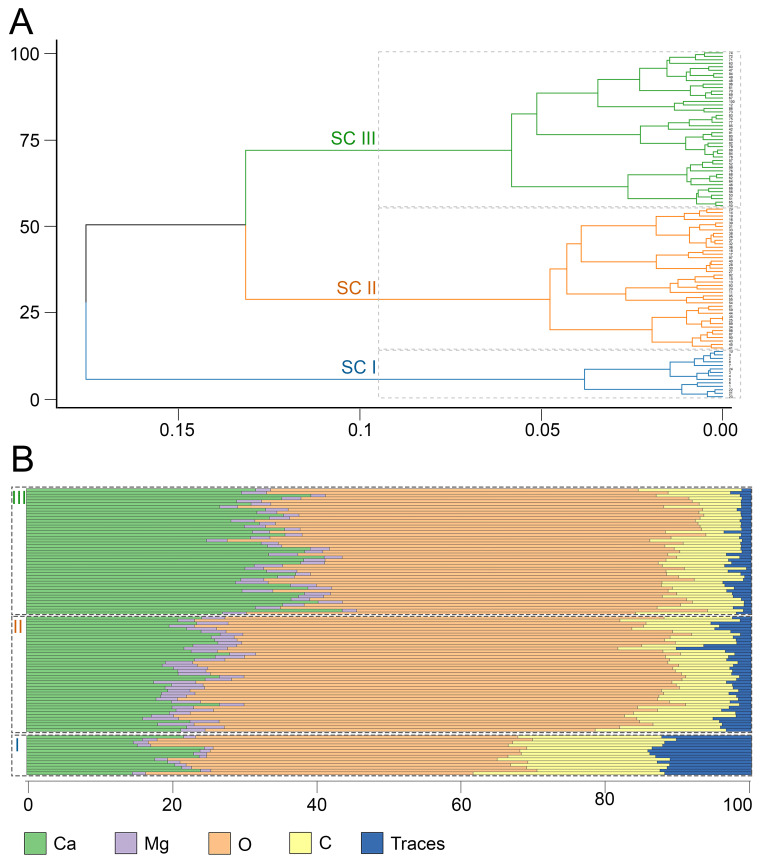
Hierarchical cluster analysis of *Onchidoris muricata* spicules in sections (SC I–III). (**A**) Dendrogram of spicule clustering. X-axis—height, Y-axis—samples. (**B**) Bar plots of the elemental composition of individual spicules in each cluster. X-axis—percentages, Y-axis—samples. Traces—amalgamation of low-represented elements (Cl, F, P).

**Figure 3 biomimetics-10-00211-f003:**
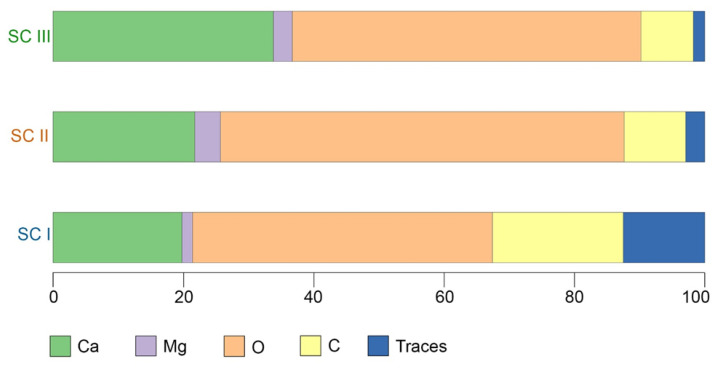
Centers of the elemental compositions (in %) of the spicule clusters in *Onchidoris muricata*. SC I–III—clusters obtained from the section data. Traces—amalgamation of low-represented elements (Cl, F, P).

**Figure 4 biomimetics-10-00211-f004:**
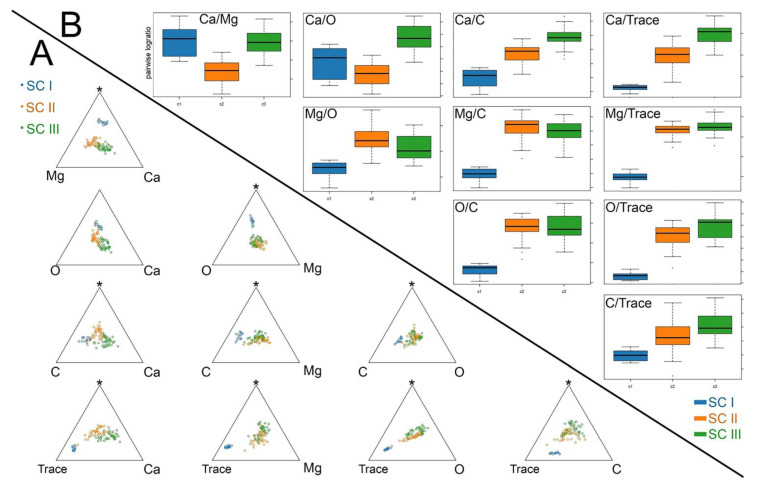
Descriptive analysis of the *Onchidoris muricata* SC I–III spicule clusters. (**A**) Matrix of centered ternary diagrams showing the distribution of spicules in the coordinates of two elements and the geometric average of the remaining elements (peak marked with an asterisk*). (**B**) Boxplots of pairwise element log ratios for each cluster. Traces—amalgamation of low-represented elements (Cl, F, P).

**Figure 5 biomimetics-10-00211-f005:**
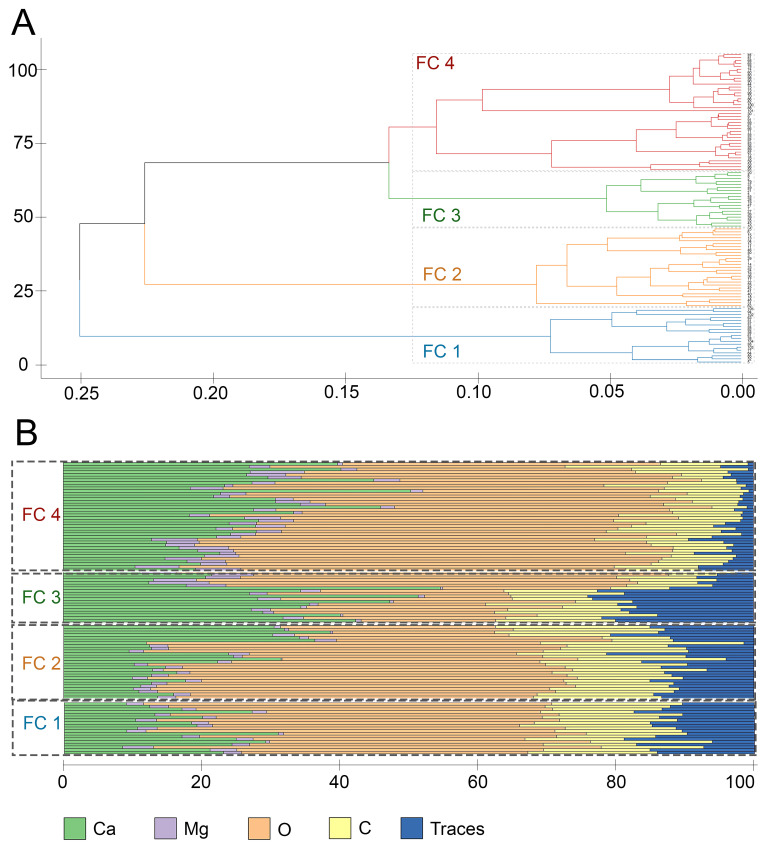
Hierarchical cluster analysis of the *Onchidoris muricata* spicules on fractures (FC I–IV). (**A**) Dendrogram of spicule clustering. X-axis—height, Y-axis—samples. (**B**) Bar plots of the elemental composition of individual spicules in each cluster. X-axis—percentages, Y-axis—samples. Traces—amalgamation of low-represented elements (Cl, F, P).

**Figure 6 biomimetics-10-00211-f006:**
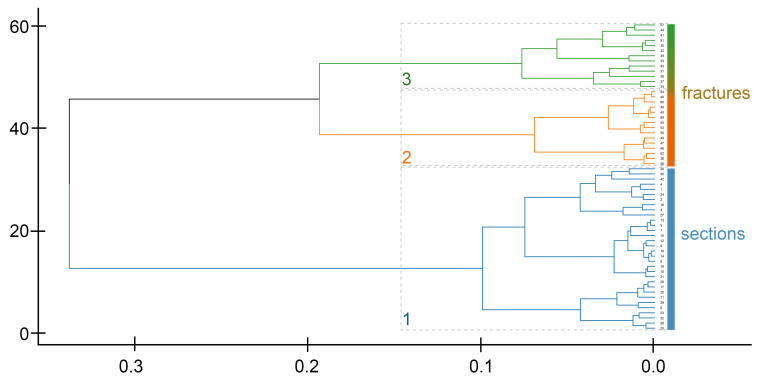
Dendrogram of hierarchical cluster analysis of the *Onchidoris muricata* spicules in both fractures and sections.

**Figure 7 biomimetics-10-00211-f007:**
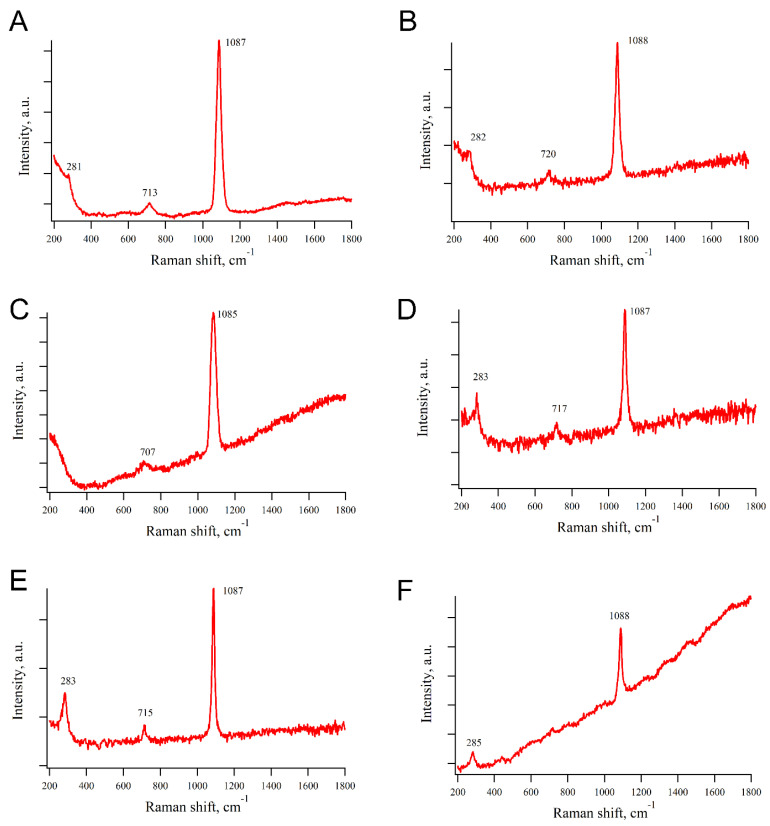
Raman spectra of the *Onchidoris muricata* spicules. (**A**) Spicules of 3 mm long *O. muricata*; (**B**,**C**) spicules of 6 mm long *O. muricata*; (**D**–**F**) spicules of 12 mm long *O. muricata*. Each panel shows only the part of the spectrum containing the characteristic peaks; no peaks occurred beyond the shown parts (Appendix A).

**Table 1 biomimetics-10-00211-t001:** Centers of the elemental compositions (in %) of the spicule clusters in *Onchidoris muricata*. SC I–III—clusters obtained from the section data; FS I–IV—clusters obtained from the fracture data. Traces—amalgamation of low-represented elements (Cl, F, P). Total variance—a measure of the global data spread.

Cluster	Number of Spicules	C	O	Ca	Mg	Traces	Total Variance
SC I	14	20.07	46.05	19.80	1.60	12.48	0.1063
SC II	41	9.47	61.96	21.72	3.94	2.9	0.2553
SC III	45	8.06	53.49	33.82	2.88	1.74	0.2567
FC I	19	14.79	29.69	36.53	1.26	17.72	0.7958
FC II	27	11.58	53.55	29.59	3.20	2.07	0.7080
FC III	19	10.46	62.46	16.75	5.32	5.00	0.3288
FC IV	40	17.31	53.32	15.91	1.91	11.54	0.9920

**Table 2 biomimetics-10-00211-t002:** Pairwise element ratios for the spicule clusters in *Onchidoris muricata*. Each cell contains the minimal ratio—mean ratio—maximal ratio of the row to the column. SC I–III—clusters obtained from the section data; FS I–IV—clusters obtained from the fracture data. Traces—amalgamation of low-represented elements (Cl, F, P).

	Ca	Mg	O	C	Traces	
Ca	1	7.13—12.36—23.54	0.29—0.43—0.57	0.55—0.99—1.42	1.21—1.59—1.89	SC I
		3.02—5.51—9.08	0.25—0.35—0.48	1.12—2.29—3.99	2.1—7.48—17.71	SC II
		6.36—11.74—21.67	0.43—0.63—0.9	1.94—4.2—8.82	7.31—19.41—45	SC III
		8.49—29.01—160.74	0.77—1.23—4.12	1.36—2.47—6.38	1.21—2.06—4.49	FC I
		3.45—9.24—56.77	0.29—0.56—1.48	1.03—2.56—9.02	4.84—14.28—79.89	FC II
		1.42—3.15—9.82	0.16—0.27—0.40	0.85—1.60—3.04	1.30—3.35—8.17	FC III
		1.91—8.32—597.5	0.13—0.3—0.73	0.40—0.92—2.12	0.86—1.38—8.48	FC IV
Mg	0.04—0.08—0.14	1	0.02—0.03—0.04	0.05—0.08—0.1	0.08—0.13—0.2	SC I
	0.11—0.18—0.33		0.04—0.06—0.11	0.14—0.42—0.78	0.58—1.36—2.19	SC II
	0.05—0.09—0.18		0.04—0.05—0.08	0.14—0.36—0.65	0.64—1.65—3.85	SC III
	0.01—0.03—0.12		0.01—0.04—0.11	0.02—0.09—0.35	0.02—0.07—0.26	FC I
	0.02—0.11—0.29		0.02—0.06—0.16	0.06—0.28—0.79	0.63—1.54—3.53	FC II
	0.10—0.32—0.71		0.04—0.09—0.14	0.23—0.51—1.00	0.27—1.06—2.04	FC III
	0.002—0.12—0.52		0.0003—0.06—0.09	0.0006—0.11—0.3	0.01—0.17—0.6	FC IV
O	1.75—2.33—3.42	23.84—28.76—41.19	1	1.71—2.29—2.69	2.98—3.69—5	SC I
	2.09—2.85—3.94	8.99—15.73—25.53		2.97—6.55—9.44	5.2—21.34—41.01	SC II
	1.11—1.58—2.35	12.—18.58—26.7		3.55—6.64—12.12	13.07—30.7—86.61	SC III
	0.24—0.81—1.3	9.23—23.58—82.79		0.99—2.01—4.52	0.52—1.68—3.52	FC I
	0.68—1.81—3.48	6.12—16.73—65.83		1.9—4.63—9.02	8.72—25.84—58.33	FC II
	2.49—3.73—6.36	7.11—11.73—24.45		4.23—5.97—8.54	6.13—12.48—26.96	FC III
	1.36—3.35—7.68	11.49—27.88—2854.5		1.93—3.08—4.43	1.81—4.62—40.49	FC IV
C	0.70—1.01—1.81	9.68—12.54—20.16	0.37—0.44—0.58	1	1.23—1.61—2.2	SC I
	0.25—0.43—0.89	1.29—2.4—7.2	0.11—0.15—0.34		0.78—3.27—10.75	SC II
	0.11—0.24—0.52	1.53—2.8—6.94	0.08—0.15—0.28		2.12—4.62—12.81	SC III
	0.16—0.4—0.74	2.86—11.75—64.22	0.22—0.5—1.01		0.46—0.83—1.72	FC I
	0.11—0.39—0.97	1.27—3.62—18.13	0.11—0.22—0.53		1.68—5.59—23.83	FC II
	0.33—0.63—1.17	1.00—2.00—4.32	0.12—0.17—0.24		1.08—2.09—4.18	FC III
	0.47—1.09—2.47	3.36—9.05—1476.0	0.22—0.32—0.52		0.61—1.5—20.97	FC IV
Traces	0.53—0.63—0.82	5.05—7.79—13.11	0.20—0.27—0.34	0.45—0.62—0.81	1	SC I
	0.06—0.13—0.48	0.46—0.74—1.73	0.02—0.05—0.19	0.09—0.31—1.29		SC II
	0.02—0.05—0.14	0.26—0.61—1.57	0.01—0.03—0.08	0.08—0.21—0.47		SC III
	0.22—0.49—0.83	3.78—14.08—40.9	0.28—0.6—1.92	0.58—1.2—2.16		FC I
	0.01—0.07—0.21	0.28—0.65—1.59	0.02—0.04—0.11	0.04—0.18—0.6		FC II
	0.12—0.3—0.77	0.49—0.94—3.71	0.04—0.08—0.16	0.24—0.48—0.93		FC III
	0.12—0.73—1.16	1.66—6.04—70.5	0.02—0.22—0.55	0.05—0.67—1.64		FC IV

**Table 3 biomimetics-10-00211-t003:** Variation matrix of the spicule clusters in *Onchidoris muricata*. SC I–III—clusters obtained from the section data; FS I–IV—clusters obtained from the fracture data. Traces—amalgamation of low-represented elements (Cl, F, P).

	Ca	Mg	O	C	
Mg	0.1419				SC I
	0.0886				SC II
	0.0819				SC III
	0.9440				FC I
	0.4438				FC II
	0.2235				FC III
	1.1959				FC IV
O	0.0621	0.0259			SC I
	0.0315	0.0554			SC II
	0.0402	0.0586			SC III
	0.2007	0.5723			FC I
	0.1529	0.2652			FC II
	0.0812	0.0863			FC III
	0.2721	0.8717			FC IV
C	0.0912	0.0408	0.0177		SC I
	0.1026	0.1568	0.0723		SC II
	0.1020	0.1176	0.1002		SC III
	0.2209	0.8347	0.2060		FC I
	0.2943	0.5389	0.2017		FC II
	0.1248	0.1246	0.0369		FC III
	0.2064	1.0989	0.0402		FC IV
Traces	0.0242	0.0680	0.0204	0.0391	SC I
	0.2225	0.1005	0.2093	0.2370	SC II
	0.2162	0.1174	0.2242	0.2252	SC III
	0.1204	0.5289	0.1899	0.1610	FC I
	0.5520	0.2449	0.3380	0.5080	FC II
	0.2588	0.2654	0.2514	0.1912	FC III
	0.2885	0.3584	0.2756	0.3522	FC IV

## Data Availability

The data that support the findings of this study are available on request from the corresponding author.

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
