# Peer review of "Mineral Composition of Skeletal Elements in Dorid Nudibranchia Onchidoris muricata (Gastropoda, Mollusca)"

_biomimetics, 2025, doi:10.3390/biomimetics10040211_

Round 1
Reviewer 1 Report
Comments and Suggestions for Authors
The manuscript provides a valuable and comprehensive analysis of the mineral composition of Onchidoris muricata spicules using Energy-Dispersive X-ray Spectrometry (EDX) and Raman spectroscopy. By comparing two preparation methods—resin-embedded sectioning and dried sample fracturing—the authors address a crucial methodological gap in biomineral analysis. The study offers fresh insights into intracellular biomineralization pathways, particularly the identification of three spicule clusters with distinct elemental ratios, supporting a hypothesis of spicule transformation stages. The work generally is well-structured, logically progressing from background to methods, results, and discussion. It is recommended that the manuscript be considered for publication after minor revisions addressing the points raised in this review.
Recommendations for Improvement:
- In the Abstract section: "Two of the three clusters have uniform layered microstructure nevertheless are showing reliable differences in element ratio." Which "..nevertheless are showing.." is somehow awkward and grammatically off. Suggesting to change into: "Two of the three clusters have a uniform layered microstructure, yet they show reliable differences in elemental ratios."
- In the 4.2 section, the phrase "Using of grinding sectioning in the current study indicates that large fully monolithic spicules appear to be absent in O. muricata." Which has "Using of" is grammatically incorrect. It can be corrected into: "The use of grinding sectioning in this study indicates that large, fully monolithic spicules are absent in O. muricata."
- In the Methods section, especially the grinding and polishing procedures (pages 6-8), is exhaustive to the point where clarity suffers. Each step is meticulously described — including the exact grit sizes and polishing compounds — which is impressive but distracts from the main goal: understanding how the methods enable better EDX analysis. We can streamline by summarizing routine steps (e.g., "Samples were progressively polished using aluminum oxide abrasives ranging from P400 to P7000 grit, followed by fine polishing with cerium oxide"). Keep the detailed description only for unique or novel steps critical to the experiment’s reproducibility, such as why cerium oxide was chosen over diamond pastes.
Comments on the Quality of English Language
The English in this manuscript is generally clear, but some sentences would benefit from more precise wording and improved flow.
Author Response
Dear reviewer,
Thank you for your time and useful suggestions. We agree with all corrections and have made appropriate changes in the manuscript.
-In the Abstract section: "Two of the three clusters have uniform layered microstructure nevertheless are showing reliable differences in element ratio." Which "..nevertheless are showing.." is somehow awkward and grammatically off. Suggesting to change into: "Two of the three clusters have a uniform layered microstructure, yet they show reliable differences in elemental ratios."
The text has been corrected.
- In the 4.2 section, the phrase "Using of grinding sectioning in the current study indicates that large fully monolithic spicules appear to be absent in O. muricata." Which has "Using of" is grammatically incorrect. It can be corrected into: "The use of grinding sectioning in this study indicates that large, fully monolithic spicules are absent in O. muricata."
The text has been corrected.
- In the Methods section, especially the grinding and polishing procedures (pages 6-8), is exhaustive to the point where clarity suffers. Each step is meticulously described — including the exact grit sizes and polishing compounds — which is impressive but distracts from the main goal: understanding how the methods enable better EDX analysis. We can streamline by summarizing routine steps (e.g., "Samples were progressively polished using aluminum oxide abrasives ranging from P400 to P7000 grit, followed by fine polishing with cerium oxide"). Keep the detailed description only for unique or novel steps critical to the experiment’s reproducibility, such as why cerium oxide was chosen over diamond pastes.
We shorted this part of methods.
Reviewer 2 Report
Comments and Suggestions for Authors
The work is devoted to improving research methods of mineralization of mollusc skeletons. The methodological work was carried out at a high level, using modern analytical equipment and with the involvement of professional specialists. In my opinion, the work deserves publication after some revision.
- Lines 143-144. If I understand this correctly, the chemical composition data were normalized to 100 %. If it is so, explanations are needed. It is also importnat to undertand the original Total values. In mineralogy, total serves as an argument that content of all elements is determined correctly.
- Table 1. Please add figure captions expaning what is "Traces" and "Total variance". Check that all data are given in the same format, i.e. X.XX (7.93, yjn 7.9312 or 7.9).
- Line 315 "1088 cm - 1(symmetric" - space missing.
- Line 317 cm-1 is missing for 281, 700.
- I didn`t get the outcome in lines 372-373. If it is supposed to mean that this minerals are absent, than it should be clearly given in the text. For example, brucite has to have O-H vibrations that are found in 3000+ cm-1 region and this region is not shown. It you state absene of that minerals, please provide (analytical) arguments in the text.
- Lines 376-377 "However, why even within one organism the composition of minerals can be variable remains unclear". I do not think that it is a correect conclusion. The difference arises from using different methods. Say, when you use local methods (such as EDX) the composition will differ in neighbouring analyses that is normal since in nature nothing is completely homogenious. And we should not forget the that during the growth of any crystal or aggregate, a huge number of processes occur, such as the accumulation of impurities at the growth front, which can be captured when the growth mode changes, and many other reasons (for example, 80 factors influencing the growth of malachite were identified, although there are many more). Therefore, discussing point variations in mineral matter relative to the environment is not entirely correct. Again, if the authors had used a macro method, such as X-ray fluorescence analysis or inductively coupled plasma mass spectrometry, where the analysis is averaged over a large amount of matter, then such variations could possibly have been discussed, but only with caution.
- Line 405. Delete dot "Sergey V. Buravkov. for".
Good luck with your work.
Author Response
Dear reviewer,
Thank you for your time and useful suggestions. We agree with all corrections and have made appropriate changes in the manuscript.
Lines 143-144. If I understand this correctly, the chemical composition data were normalized to 100 %. If it is so, explanations are needed. It is also importnat to undertand the original Total values. In mineralogy, total serves as an argument that content of all elements is determined correctly.
EDX quantitative analysis is based on the conversion of the relative intensities of the characteristics of X-rays (which are counts of electrons with a specific energy in keV) through the procedure of quantitative correction (we have used ZAF correction). In fact, the relative quantities of elements are calculated as an approximation of the obtained spectrum to the theoretical one for a mixture of the identified elements. The normalization to 100% is the internal part of this mathematical procedure.
We have corrected the subsection ‘EDX analysis’ in the section ‘Materials and methods’ to make the description of quantitative analysis clearer for readers.
Table 1. Please add figure captions expaning what is "Traces" and "Total variance". Check that all data are given in the same format, i.e. X.XX (7.93, yjn 7.9312 or 7.9).
Information has added to all table, figure and supplementary material captures:
Traces – amalgamation of low-represented elements (Cl, F, P). Total variance - measure of the global data spread.
All values have been checked and corrected through the manuscript.
Line 315 "1088 cm - 1(symmetric" - space missing.
The text has been corrected.
Line 317 cm-1 is missing for 281, 700.
The text has been corrected.
I didn`t get the outcome in lines 372-373. If it is supposed to mean that this minerals are absent, than it should be clearly given in the text. For example, brucite has to have O-H vibrations that are found in 3000+ cm-1 region and this region is not shown. It you state absene of that minerals, please provide (analytical) arguments in the text.
Lines 376-377 "However, why even within one organism the composition of minerals can be variable remains unclear". I do not think that it is a correect conclusion. The difference arises from using different methods. Say, when you use local methods (such as EDX) the composition will differ in neighbouring analyses that is normal since in nature nothing is completely homogenious. And we should not forget the that during the growth of any crystal or aggregate, a huge number of processes occur, such as the accumulation of impurities at the growth front, which can be captured when the growth mode changes, and many other reasons (for example, 80 factors influencing the growth of malachite were identified, although there are many more). Therefore, discussing point variations in mineral matter relative to the environment is not entirely correct. Again, if the authors had used a macro method, such as X-ray fluorescence analysis or inductively coupled plasma mass spectrometry, where the analysis is averaged over a large amount of matter, then such variations could possibly have been discussed, but only with caution.
This part of the Discussion (Lines 372-377) have been rewritten to clarify our argumentation. Also we added figure to the supplementary materials (figure S1) with full spectra showing than no peaks occur beyond the spectrum regions in fig 7.
Line 405. Delete dot "Sergey V. Buravkov. for".
The text has been corrected.